# Comprehensive Exonic Sequencing of Known Ataxia Genes in Episodic Ataxia

**DOI:** 10.3390/biomedicines8050134

**Published:** 2020-05-25

**Authors:** Neven Maksemous, Heidi G. Sutherland, Robert A. Smith, Larisa M. Haupt, Lyn R. Griffiths

**Affiliations:** Genomics Research Centre, Institute of Health and Biomedical Innovation (IHBI), School of Biomedical Sciences, Q Block, Queensland University of Technology (QUT), 60 Musk Ave, Kelvin Grove Campus, Brisbane, Queensland 4059, Australia; n.maksemous@qut.edu.au (N.M.); heidi.sutherland@qut.edu.au (H.G.S.); r157.smith@qut.edu.au (R.A.S.); larisa.haupt@qut.edu.au (L.M.H.)

**Keywords:** episodic ataxia, whole exome sequencing, acetazolamide

## Abstract

Episodic Ataxias (EAs) are a small group (EA1–EA8) of complex neurological conditions that manifest as incidents of poor balance and coordination. Diagnostic testing cannot always find causative variants for the phenotype, however, and this along with the recently proposed EA type 9 (EA9), suggest that more EA genes are yet to be discovered. We previously identified disease-causing mutations in the *CACNA1A* gene in 48% (*n* = 15) of 31 patients with a suspected clinical diagnosis of EA2, and referred to our laboratory for *CACNA1A* gene testing, leaving 52% of these cases (*n* = 16) with no molecular diagnosis. In this study, whole exome sequencing (WES) was performed on 16 patients who tested negative for *CACNA1A* mutations. Tiered analysis of WES data was performed to first explore (Tier-1) the ataxia and ataxia-associated genes (*n* = 170) available in the literature and databases for comprehensive EA molecular genetic testing; we then investigated 353 ion channel genes (Tier-2). Known and potential causal variants were identified in *n* = 8/16 (50%) patients in 8 genes (*SCN2A*, p.Val1325Phe; *ATP1A3*, p.Arg756His; *PEX7*, p.Tyr40Ter; and *KCNA1*, p.Arg167Met; *CLCN1*, p.Gly945ArgfsX39; *CACNA1E*, p.Ile614Val; *SCN1B*, p.Cys121Trp; and *SCN9A,* p.Tyr1217Ter). These results suggest that mutations in these genes might cause an ataxia phenotype or that combinations of more than one mutation contribute to ataxia disorders.

## 1. Introduction

Episodic ataxias are autosomal-dominant neurological disorders characterized by severe episodes of ataxia (discoordination), variably associated with progressive ataxia and ictal/interictal features. Currently, there are eight recognized forms of episodic ataxia (EA1–EA8) [1], with a ninth recently suggested subtype (EA9) linked to *FGF14* [2]. EA2 and EA1 are the most commonly reported hereditary episodic ataxias. EA1, which features persistent myokymia and usually brief ataxia episodes, is caused by mutations in the *KCNA1* gene, encoding the neuronal potassium channel Kv1.1 [3]. EA2 episodes are characterized by intermittent attacks of ataxia that are often accompanied by nystagmus, vertigo, and muscle weakness [4]. The attacks, which last from hours to days, can be completely relieved in many patients through acetazolamide treatment. In the last two decades a large variety of loss of function mutations in *CACNA1A*, which encodes the voltage-gated calcium channel alpha1a subunit (Ca_v2.1_), were found in patients with Episodic Ataxia type 2 (EA2) [5]. The genetic complexity and the overlapping phenotypes of ataxias led our group to screen the *CACNA1A* gene in 31 cases that were clinically diagnosed with suspected EA2 and were referred to our laboratory for genetic testing. Using a targeted next generation sequencing (NGS) gene panel, including full exonic coverage of *CACNA1A*, we identified causal mutations in 15 of these cases (48%) [6]. Despite this success, 16 cases remained with no molecular genetic diagnosis [6,7]. This might be due to the fact that mutations in the known EA genes account for less than 50% of the suspected EA cases, suggesting genetic heterogeneity and that more ataxia genes are yet to be identified [1,6,7].

Episodic and non-episodic (congenital, progressive, nonprogressive) ataxias are a highly genetically and clinically heterogeneous group of neurological disorders that affect individuals of all age groups. The less common EA types were often found in individual families, and might be the result of family-specific mutations with unique effects. Indeed, not all EA types actually have a causative gene identified, with *CACNB4* and *SLC1A3* causing EA5 and EA6, respectively, along with a possible candidate of *UBR4* for EA8 [8,9]. The remaining EA types are only linked to chromosomal regions, with EA3 linked to 1q42, EA7 to 19q13, and EA4 not linked to known loci [8,10]. In the absence of distinguishing clinical features and the potential involvement of many genes other than the ion channel genes [8,11], a comprehensive molecular genetic test is essential.

Whole exome sequencing (WES) is a beneficial method that has led to the identification of causal gene mutations in various neurological disorders, including ataxias [7,12,13]. In this study, we investigated the EA2 cases (*n* = 16) found to be negative for exonic *CACNA1A* mutations in our previous study [6], by performing WES with subsequent comprehensive tier-targeted analysis of ataxia and ion channel genes. We report the identification of known or potentially causal variants in 8 different genes in 8 of the 16 patients (50%). Our results suggest that an EA2 phenotype might result from a diverse spectrum of variants and that WES analysis of patients can provide an increased diagnostic yield.

## 2. Materials and Methods

### 2.1. Cohort Summary

In total, 16 EA2 subjects (9 Females and 7 Males) previously screened to be negative for *CACNA1A* gene mutations, underwent whole exome sequencing (WES) (Table 1).

The study was conducted in accordance with the Declaration of Helsinki, and the protocol was approved by the Human Research Ethics Committee of the Queensland University of Technology (approval number: 1800000611, 02/11/2018).

### 2.2. Whole Exome Sequencing Pipeline

Whole Exome Sequencing (WES) was performed on isolated genomic DNA of the 16 participants with a suspected clinical diagnosis of EA2. All participants were previously referred for molecular genetic diagnostics testing at the accredited Genomics Research Centre (GRC) laboratory. WES libraries were prepared using the previously extracted genomic DNA and Ion AmpliSeq Exome RDY library preparation kit (Thermo Fisher Scientific, Carlsbad, CA, USA), according to the manufacturer’s protocol. The 16 libraries were sequenced on the Ion Proton sequencer, using 8 Ion P1 Chips (Thermo Fisher Scientific, Carlsbad, Ca., USA). Sequence reads were aligned to the human reference genome (hg19), the variants were called and the VCF files were generated using the Ion Torrent Suite software v5.10.1 (Carlsbad, Ca., USA). An average coverage depth of 140× was achieved. The VCF files were then uploaded in the locally hosted Ion Reporter Server software 5.10 (Thermo Fisher Scientific, Carlsbad, Ca., USA) for automated variant annotation and filtering. The bam format files generated by the Torrent Suite v5.10.1 were uploaded into the Integrative Genomics Viewer (IGV v2.3) software (http://www.broadinstitute.org/igv), to visually examine the variants, before Sanger sequencing validation.

### 2.3. Variant Filtering Workflow

To select potential variants, iterative filtering was performed to focus on variants that alter the protein-coding regions, indels, and canonical splice-sites in 170 ataxia and ataxia-associated genes, as a first tier. Then, the variants were filtered to cover 353 ion channel genes as a second-tier analysis (Figure 1). Additional filtering to remove all variants found in the databases dbSNP (www.ncbi.nlm.nih.gov/projects/SNP/Build138), 1000 Genomes Project (www.1000genomes.org), and gnomAD (http://gnomad-old.broadinstitute.org/) at > 0.0001 minor allele frequency (MAF). The remaining variants were then visually confirmed or rejected by the IGV software to remove any platform or low coverage artefacts before the Sanger sequencing validation.

### 2.4. Sanger Sequencing (SS) Method

Following the variant filtering process, integrated DNA Technologies software (https://sg.idtdna.com) was used to design the 8 primer pairs (sense and antisense) for the 8 identified pathogenic and likely pathogenic variants. Primers used in this study and the Polymerase Chain Reaction (PCR) protocols are available upon request. The PCR products were then purified using the Affymetrix ExoSAP-IT reagent, labelled using ABI BigDye terminator (BDT) v3.1, purified and concentrated using ethanol precipitation, according to the previously described protocol [14]. Each sample was sequenced in both directions (sense and antisense) on an Applied Biosystems 3500 Series Genetic Analyzer (Thermo Fisher Scientific, Scoresby, Victoria, Australia). Sequencing traces were analyzed using the Chromas 2.33 software and was compared to the WES results and the NCBI reference sequences. The variants were reported according to the Human Genome Variation Society (HGVS) nomenclature.

## 3. Results

WES was performed in 16 individuals with ataxic symptoms (Table 1), who had been referred by clinicians for EA2 testing, but were found to be negative for pathogenic *CACNA1A* mutations, in our previous study [6]. A total of 102291 genetic variants in 19083 genes were identified in the 16 individuals. For comprehensive EA molecular genetic testing, we performed WES along with a tiered analysis approach, to explore the variants in ataxia and ataxia-associated genes (*n* = 170, Tier-1) from the current literature and the Pubmed database (https://www.ncbi.nlm.nih.gov/pubmed/) (Appendix A). Due to the crucial role of the ion channel genes in the development and propagation of action potentials in neurons and muscle cells, 353 ion channel genes were additionally investigated for potentially pathogenic variants (Tier-2) (Appendix A). The two-tiers of analysis were performed through Ion Reporter v5.10 filtering, as summarized in Figure 1. Of the total variants identified in the 16 individuals, 1457 variations (exonic, intronic, canonical splice site, and indels) were identified in the 170 ataxia and ataxia associated Tier-1 gene list. All variants affecting amino-acid composition of the protein with minor allele frequency (MAF) < 0.0001 were filtered (*n* = 38/1457); as listed in Appendix A. Of these 38 variants identified in the initial analysis, the only variants classified pathogenic or likely pathogenic by the American College of Medical Genetics and Genomics standards and guidelines (ACMG) are discussed in [15]. In total, 4 pathogenic and likely pathogenic variants in 4 genes were identified in Tier-1 analysis and all 4 variants were validated by Sanger Sequencing (Appendix A). Tier-2 analysis identified a total of 1799 variations (exonic, intronic, canonical splice site, and indels) in 353 ion channel genes. After filtering for all amino-acid changing variants with MAF <0.0001, 28 candidate variants were identified (Appendix A). The 28 variants were interpreted according to the ACMG guidelines and, here, we report 7 pathogenic or likely pathogenic variants in 7 genes (Table 2), all validated by the Sanger sequencing (Appendix A). Of these, 3 variants (p.Val1325Phe in *SCN2A*, p.Arg756His in *ATP1A3*, and p.Arg167Met in *KCNA1*) overlap with those identified from the analysis of the ataxia-related genes identified in Tier-1.

Schematic representation of the variant filtering. Tier-1 analysis (left)—total variant counts at each stage of filtering using the Ion Reporter software resulted in four non-synonymous pathogenic and likely pathogenic mutations in 170 ataxia-gene panel. Tier-2 analysis (right)—total variant counts at each stage of filtering resulted in 7 pathogenic and likely pathogenic variants in 353 ion channel genes. All of these were confirmed by the Sanger sequencing method.

### 3.1. Genetic Analysis

Genetic analysis for the previously undiagnosed 16 EA cases identified 8 pathogenic or likely pathogenic mutations (*n* = 8/16; 50%) in 8 genes (*SCN2A*, p.Val1325Phe; *ATP1A3*, p.Arg756His; *PEX7*, p.Tyr40Ter; and *KCNA1*, p.Arg167Met; *CLCN1*, p.Gly945ArgfsX39; *CACNA1E*, p.Ile614Val; *SCN1B*, p.Cys121Trp; and *SCN9A*, p.Tyr1217Ter) (Table 2). Additional variants were interpreted by the ACMG guidelines as variants of uncertain significance, likely benign, and benign, are listed in Appendix A and were not assessed any further.

### 3.2. Ataxia Gene-Panel Analysis (Tier-1)

Using the ataxia gene-panel list comprised of 170 genes; we identified a total of 38 novel or very rare variants. Herein, we only report the four heterozygous variants classified to be pathogenic and likely pathogenic, according to the ACMG guidelines (Table 2). All four variants were previously reported to cause—episodic ataxia (EA) in *SCN2A*; alternating hemiplegia of childhood in *ATP1A3*; Refsum disease (RD) in *PEX7*; and episodic ataxia type 1 (EA1) in *KCNA1*. In detail:

**Case-2:** The father of the index case of a three-generation family found to harbor a de novo missense variant was predicted to be likely pathogenic in *SCN2A* (NM_001040143.1 c. [3973G > T]; [3973G=] p.[Val1325Phe]; [Val1325=]); these data were published separately [7].

**Case-6:** Presented at the age of 6 with episodes of ataxia associated with fever; gait ataxia, and past pointing. He experienced 4–5 attacks per year that lasted for several days. Acetazolamide treatment was not effective. Genetic analysis identified a known missense pathogenic mutation in *ATP1A3* (NM_152296.4 c. [2267G > A]; [2267G=] p.[Arg756His]; [Arg756=]) [16].

**Case-19:** Presented at the age of 38 with Vertigo, fluctuating ataxia, and abnormal nerve excitability. She was clinically diagnosed with suspected EA2 and tested negative for *CACNA1A* gene mutations. Genetic analysis using WES identified a known nonsense pathogenic mutation in *PEX7* exon1 (NM_000288.3 c. [120C > G]; [120C=] p.[Tyr40Ter]; [Tyr40=]) [17,18].

**Case-23:** Presented at the age of 24 with unsteadiness and muscle myokymia, triggered by physical exercise. He reported a partial improvement with acetazolamide. Genetic analysis identified a known missense mutation in *KCNA1* (NM_000217.2 c. [500G > T]; [500G=] p.[Arg167Met]; [Arg167=]). The mutation was located in the N-terminus of this ion channel, at its junction with the transmembrane S1 domain [19].

### 3.3. Ion Channel Analysis (Tier-2)

A total of 28 novel or rare variants were identified in the Tier-2 analysis of 353 ion channel genes (Appendix A). Here, we report the four pathogenic and likely pathogenic variants identified in seven EA cases in *CLCN1*, *CACNA1E*, *SCN1B*, and *SCN9A*, in addition to the three variants in *ATP1A3*, *SCN2A*, and *KCNA1* listed in the Tier-1 analysis.

**Case-14:** Presented at the age of 27 with severe ataxia, nausea, vomiting, and nystagmus. She reported 4 attacks/year and acetazolamide dramatically relieved her symptoms. WES analysis revealed a pathogenic frameshift insertion in *CLCN1* (NM_000083.2: c.[2831dupC] p.[(Gly945fs)]; [Gly945=]).

**Case-21:** This patient was referred for *CACNA1A* genetic testing at the age of 80. He was clinically diagnosed with late onset of episodic ataxia at the age of 60. Genetic analysis using WES revealed a likely pathogenic missense variant in *CACNA1E* (NM_001205293.1: c.[1840A > G]; [1840A=] p.[(Ile614Val)]; [Ile614=]).

**Case-25:** Presented at the age of 59 with episodes of ataxia and no migraine headache. Family histories of similar symptoms were reported in his father, paternal aunt, grandmother, and sisters. Genetic analysis identified a missense variant in the *SCN1B* gene (NM_199037.4: c.[363C > G]; [363C=] p.[Cys121Trp]; [Cys121=]), previously known to cause Febrile seizures and generalized epilepsy [20] and classified to be likely pathogenic, according to the ACMG guidelines.

**Case-31:** Presented at the age of 56 with progressive ataxia and vertigo. She had no family history of similar phenotypes. Genetic analysis identified a novel nonsense variant in *SCN9A* (NM_002977.3: c.[3651T > G];[3651T=] p.[(Tyr1217Ter)]; [Tyr1217=]).

## 4. Discussion

Due to the complexity of clinical manifestations and the significant phenotypic overlap, clinical diagnosis of EA is challenging. Therefore, a comprehensive genetic analysis, including the use of whole exome sequencing (WES) provides the opportunity to confirm clinical diagnoses and identify the specific type of ataxia in the cases caused by known genes, and also to identify new and novel genes and variants that might be involved in this family of disorders.

In this study, the 16 cases that were negative for pathogenic *CACNA1A* mutations underwent WES followed by a tiered analysis approach, which identified causal mutations in genes previously implicated in various neurological disorders with symptomatic overlap with ataxia and epilepsies [7,12]. Using the ataxia gene-panel (Tier-1), we identified 4 causal variants in 4 cases (25%), while targeted analysis using the ion channel gene-panel list (Tier-2) identified a total of 7 variants (44%), of which 3 variants were also identified in Tier-1. In total, 8 pathogenic and likely pathogenic variants were identified in the 8/16 cases (50% diagnostic yield) (Table 2).

### 4.1. Ataxia Gene-Panel Variants (Tier-1)

We identified four pathogenic and likely pathogenic mutations in 4 genes (*SCN2A*, p.Val1325Phe; *ATP1A3*, p.Arg756His; *PEX7*, p.Tyr40Ter; and *KCNA1*, p.Arg167Met) likely to contribute to the EA phenotype in 4 patients (Case-2, Case-6, Case-19, and Case-23), respectively. All four mutations were previously reported [7,16,17,19].

Pathogenic variants in at least 21 patients with *SCN2A*-associated episodic ataxia were recently reported, including our Case-2 (p.Val1325Phe), with age of onset ranging from early infancy (10 months) to adolescent onset (14 years) and with wide phenotypic variation [7,21]. *SCN2A* EA treatment options remain unsatisfactory, ranging from the limited effect of acetazolamide in the cases that harbor a de novo missense variant (data previously published) [7] to a dramatic relief in patients who harbor a loss of function (nonsense) variants, highlighting the importance of the variant effect and location, in the treatment strategy.

The *ATP1A3* c. 2267G > A (p.Arg756His) variant we identified was previously reported in patients with ataxia, hypotonia, and seizures [16,22], with the age of onset ranging between 9 months to 3 years. To date, including Case-6, 13 patients with mutations at the Arg-756 were reported, with symptoms including fever-induced paroxysmal weakness and encephalopathy (FIPWE). Gait ataxia accompanied with migraine is also a predominant feature in adults with p.Arg756His and flunarizine treatment, a calcium-channel blocker, which has been reported to be effective [16].

Mutations in the *PEX7* gene result in a broad clinical spectrum of effects, including muscle weakness, and ataxia [18]. The p.Tyr40Ter stop-gain variant, introduces a premature stop codon in the N-terminal region of the peroxin 7 protein, and was previously reported in 2 patients diagnosed with rhizomelic chondrodysplasia punctate (RCDP) in whom a second variant was not identified, and in 2 individuals with compound heterozygosity *PEX7* variants, who were diagnosed with the Refsum disease [17,18]. Ataxia was a prominent feature in the latter probands, showing a late age of onset, ranging between 12 (proband-1) and 34 (proband-2 brother) years. Van den Brink and colleagues (2003) demonstrated that the variant led to a deficiency of phytanic acid alpha-oxidation, PhyH activity, and plasmalogen synthesis, defective subcellular localization, and an inability to restore the thiolase protein transport [18]. While the Refsum disease is usually a recessive disorder, it was demonstrated that there is a detectable loss of phytanic acid oxidation in tissues derived from unaffected parents of the disease sufferers. As such, it is possible that other factors in our case have exacerbated the effect of a heterozygous mutation, sufficient to cause late onset ataxia [23]. The link between this nonsense variant and the ataxia with vertigo phenotype, as in our case, needs more investigation.

Heterozygous mutations in the ion channel *KCNA1* cause episodic ataxia type 1 (EA1), a disorder characterized by brief episodes of ataxia (seconds to minutes), accompanied by involuntary muscle movement (Myokymia). The *KCNA1* c.500G > T (p.Arg167Met) mutation was previously described in an individual with EA1 [19]. The mutation cause a loss of potassium channel (K_v_1.1) activity, when expressed in human embryonic kidney cells [19]. Unusual clinical manifestations such as muscle weakness during an attack and deafness were also described in individuals carrying the p.Arg167Met variant, however, this was not reported for our Case-23, who had symptoms typical for EA1. Treatment with acetazolamide had a partial effect, in our case, and this might be due to the functional loss nature of the mutation.

### 4.2. Ion Channel Gene-Panel Variants (Tier-2)

Ion channel malfunctions in *CACNA1*, *SCN*, *KCN*, and *CLCN* genes were associated with a variety of neurological disorders such as episodic ataxia, spinocerebellar ataxia, and epilepsy. To explore the potential contribution of the genetic variation to the ataxia phenotype in the ion channel genes, we also analyzed 353 ion channel genes (Tier-2) in the 16 EA patients who tested negative for *CACNA1A* mutations. A total of 7 variants (*n* = 7/16, 44%) were identified by Tier-2 analysis, in 7 genes across 7 EA patients. In addition to the variants identified in *SCN2A*, *ATP1A3*, and *KCNA1* in the ataxia-gene panel discussed above, 4 highly suggestive candidate variants (*n* = 4/7) in the *CLCN1*, p.Gly945ArgfsX39; *CACNA1E*, p.Ile614Val; *SCN1B*, p.Cys121Trp; and *SCN9A*, p.Tyr1217Ter genes were identified.

Loss-of-function mutations of the chloride channel ClC-1 encoded by *CLCN1*, causing hyperexcitability in the skeletal muscle are commonly found in patients with myotonia, an impairment of muscle relaxation. Although *CLCN1* is expressed in the brain and mutations have been reported in patients with epilepsy [24], the functional role of *CLCN1* mutations in other neurologic phenotypes including ataxia remains unclear. The frameshift insertion in Case-14 (c.2831dup) results in a C nucleotide duplication, creating a premature stop codon at position 39 of the new reading frame, denoted p.Gly945ArgfsX39. The variant was previously reported in an individual with non-dystrophic myotonia who also carried a second *CLCN1* pathogenic variant [25]. In addition to muscle stiffness, the major clinical manifestation of non-dystrophic myotonia, common symptoms such as pain, weakness, and fatigue were also reported [26]. The effect of a *CLCN1* frameshift in the acetazolamide-responsive patient (Case-14) with severe ataxia, nausea, vomiting, and nystagmus at the age of 27, raised the possibility of phenotypic variability within the same genotype p.Gly945ArgfsX39 mutation. However, a p.Glu131Lys variant in the potassium channel (*KCNS2*), classified as a variant of unknown significance (VOUS) (Appendix A) in this patient might also contribute to pathogenicity, and thus requires further functional validation.

Mutations in *CACNA1E*, *SCN1B*, and *SCN9A* were previously associated with various epilepsy syndromes. Three likely pathogenic variants in these genes were identified in Case-21 (*CACNA1E* p.Ile614Val), Case-25 (*SCN1B* p.Cys121Trp), and Case-31 (*SCN9A* p.Tyr1217Ter), with very late onset of ataxia at the age of 60, 59, and 56, respectively. *CACNA1E* is located on chromosome 1q25.3 which encodes the α1-subunit of the voltage-gated Ca_V_2.3 channel and is highly expressed in the central nervous system. A recent study has revealed a wide range of clinical phenotypes linked to *CACNA1E*, including developmental and epileptic encephalopathy with contractures, macrocephaly, and dyskinesias [27]. The *CACNA1E* gene was also identified as a candidate gene for neurodevelopmental disorders [28]. Helbig and colleagues reported a gain of function effect of all missense variants identified in the cytoplasmic parts of all four S6 transmembrane segments of *CACNA1E* [27]. However, the functional effect of the variant identified in this study and located in the domain II S5 transmembrane region needs more investigation.

The *SCN1B* c. 363C > G (p.Cys121Trp) mutation in Case-25 was previously described in a large family of 378 individuals from 6 generations, with febrile seizures and generalized epilepsy (GEFS) [20]. The mutation was believed to disrupt the disulfide bridge between two highly conserved Cysteine residues, including Cys121 in *SCN1B*, which might alter the secondary structure of the extracellular domain [20]. Functional analysis using two-electrode voltage clamp recording has shown that the mutation interferes with the ability of the β1 subunit to modulate sodium channel gating, leading to a loss of function that might cause persistent inward Na+ currents in neurons, resulting in a more depolarized membrane potential and hyperexcitability [20]. Chen and colleagues reported that SCN1B null mice resulted in a dramatic phenotype that includes an ataxic gait and spontaneous seizures [29]. Late onset ataxia was not previously reported in patients with *CACNA1E* and *SCN1B* mutations and it is possible that the phenotype might be affected by other genetic variants or environmental influences. Further investigation and follow-up of the effect of the identified variants in later life are required.

The *SCN9A* c.3651T > G (p.Tyr1217Ter) nonsense mutation, located in the extracellular domain III S1-S2 in Case-31 was not present in any publication or databases, including gnomAD (https://gnomad.broadinstitute.org/) and dbSNP (https://www.ncbi.nlm.nih.gov/SNP/), at the time of writing. The patient presented with late onset progressive ataxia and vertigo. This variant was classified as a pathogenic variant, according to the ACMG guidelines. *SCN9A,* encodes the Na_v_1.7 ion channel, and is expressed primarily in neurons of the dorsal root ganglia, as well as in the brain [30,31]. Mutations in *SCN9A* were reported in three disorders, autosomal dominant primary erythermalgia (PE), paroxysmal extreme pain disorder (PEPD), and autosomal recessive channelopathy-associated insensitivity to pain (CIP). The association of *SCN9A* with epilepsy and other PE, PEPD, CIP phenotypes resembles the role of *CACNA1A* in causing discrete phenotypes such as epilepsy, familial hemiplegic migraine, and spinocerebellar ataxia and episodic ataxia. Additional work to establish the functional effects of our identified mutation in *SCN9A* and to definitively link this variant to the ataxia phenotype is still required.

Last, we identified a number of variants of unknown significance in EA cases who did not harbor any pathogenic or likely pathogenic gene mutations in the two tiers lists examined. For instance, Case-4 harbors a novel missense variant (c.1567G > A p.Asp523Asn) in the *SCN10A* gene, and Case-20 harbors a very rare variant (c.3893T > C p. Met1298Thr) in the same gene. The *SCN10A* gene encodes the voltage-gated sodium channel alpha subunit and mutations in *SCN10A* cause Episodic Pain Syndrome, Familial 2, and Sodium Channelopathy-Related Small Fiber Neuropathy. Further investigation is needed to uncover the molecular mechanisms of these variants and how they might contribute to ataxia phenotypes.

## 5. Conclusions

In summary, WES was undertaken in 16 clinically diagnosed, with suspected EA2, patients who tested negative for pathogenic mutations in *CACNA1A*, the gene most commonly causative of EA type 2. Tiered analysis of the WES data was performed and 8 known and potentially causal variants *n* = 8/16 (50%) were identified in 8 genes (*SCN2A*, p.Val1325Phe; *ATP1A3*, p.Arg756His; *PEX7*, p.Tyr40Ter; and *KCNA1*, p.Arg167Met; *CLCN1*, p.Gly945ArgfsX39; *CACNA1E*, p.Ile614Val; *SCN1B*, p.Cys121Trp; and *SCN9A,* p.Tyr1217Ter). Molecular diagnosis remains unknown in 50% of cases (*n* = 8/16). These results suggest that the genes we have identified in this research might cause overlapping phenotypes, which include ataxia, or that a combination of more than one mutation might be implicated in the ataxia disorder in these patients. Further extensive work, including biochemical assays, segregation, and functional studies of variants (including the variants classified as VOUS) detected in this and others studies are needed to give a more detailed understanding and accurate classification for these variants in disease pathophysiology.

Additionally, the cases with no detectable mutation identified in the candidate gene sets suggest that there might be other ataxia-phenotype-causing genes that are yet to be discovered. Additionally, it is possible that non-genetic or genetic and environmental interaction causes might be at work in some of our patients. All patients were identified as fitting the EA2 criteria well-enough to warrant *CACNA1A* testing by their referring neurologist. Despite this, injury, infection, auto-immune diseases, and similar conditions causing EA-like symptoms might have escaped clinical detection, especially if they were relatively mild and were acting in a genetic context more sensitive to ataxia-related phenotypes. Improvements in genetic testing, clinical testing, and detailed descriptions of the clinical phenotypes will likely lead to earlier and more accurate diagnosis of these disorders, and to more effective treatment strategies. Shared, well-curated international ataxia databases are also necessary as more patients and variants/mutations are identified.

## Figures and Tables

**Figure 1 biomedicines-08-00134-f001:**
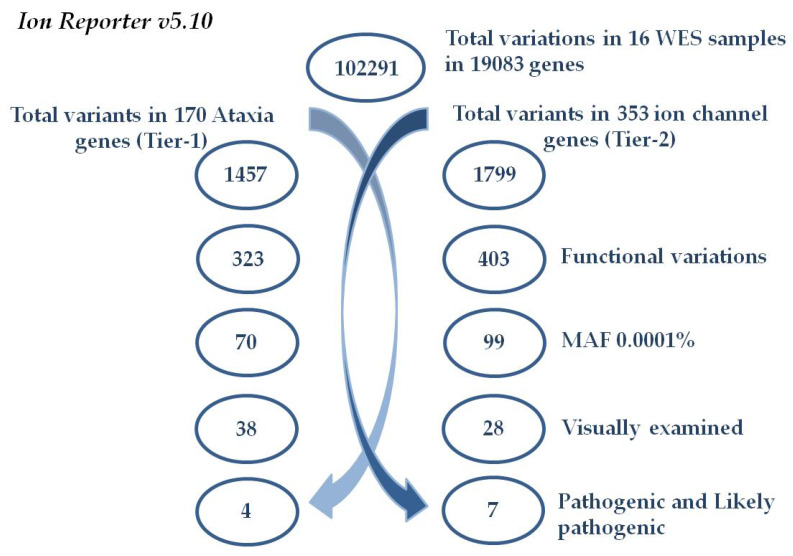
Variant filtering workflow using Ion Reporter v5.10.

**Table 1 biomedicines-08-00134-t001:** The available clinical features of the 16 episodic ataxia patients.

ID	Age at Test Request (Years)	Gender	Age at Onset (Years)	Familial History	Clinical Information	Acetazolamide Response
1	2.5	F	2	NO	Episodes of ataxia	-
2	15	M	-	YES	Episodes of ataxia associated with fever and gait ataxia and past pointing. Attacks triggered by a mild head trauma	Positive
4	37	F	-	-	Episodes of ataxia, possible hemiplegic migraine	-
6	6	M	1	YES	Episodes of ataxia associated with fever; gait ataxia; 4–5 attacks/year that can last for several days	Negative
7	60	M	-	-	Episodes of ataxia	-
10	37	F	-	-	Daily attack	-
11	26	M	-	-	Episodes of ataxia	-
12	76	F	61	YES	Attacks of vertigo, +/− headache every few months	Partial
14	27	F	-	-	Severe ataxia, nausea, vomiting, nystagmus; Four attacks/year	Positive
16	42	F	-	YES	Episodic ataxia	-
19	38	F	-	-	Vertigo, fluctuating ataxia, abnormal nerve excitability	-
20	80	F	-	-	Episodes of ataxia	-
21	80	M	60	-	Late onset of episodic ataxia	-
23	24	M	-	-	Unsteadiness, muscle myokemia, exercise induced	Partial
25	59	M	-	YES	Ataxia, no headache.	-
31	56	F	-	NO	Progressive ataxia, vertigo	-

Abbreviations: F: Female; M: Male; (-): Information not available.

**Table 2 biomedicines-08-00134-t002:** Heterozygous mutations identified in episodic ataxia cases identified by whole exome sequencing.

ID	Locus (hg19)	Ref	Genes	Transcript	Amino Acid Change	Coding	GnomAD Frequency	ACMG Rules	Verdict
2	chr2:166231195	G	*SCN2A*	NM_001040143.1	p.Val1325Phe	c.3973G > T	-	PM1,PM2,PP2,PP3	Likely Pathogenic
6	chr19:42474691	C	*ATP1A3*	NM_152296.4	p.Arg756His	c.2267G > A	-	PM1,PM2,PM5,PP2,PP3,PP5	Pathogenic
14	chr7:143048918	G	*CLCN1*	NM_000083.2	p.Gly945fs	c.2831dup	0.000016	PVS1,PM2,PP3	Pathogenic
19	chr6:137143923	C	*PEX7*	NM_000288.3	p.Tyr40Ter	c.120C > G	0.0000907	PVS1, PS3, PM2, PP3.	Pathogenic
21	chr1:181689430	A	*CACNA1E*	NM_001205293.1	p.Ile614Val	c.1840A > G	0.00000804	PM1,PM2,PP2,PP3	Likely Pathogenic
23	chr12:5021044	G	*KCNA1*	NM_000217.2	p.Arg167Met	c.500G > T	-	PM1,PM2,PP2,PP3	Likely Pathogenic
25	chr19:35524558	C	*SCN1B*	NM_199037.4	p.Cys121Trp	c.363C > G	0.0000141	PM1,PM2,PP3,PP5	Likely Pathogenic
31	chr2:167094721	A	*SCN9A*	NM_002977.3	p.Tyr1217Ter	c.3651T > G	-	PVS1,PM2,PP3	Pathogenic

Hg19: human genome 19; Ref: reference allele; ACMG: American College of Medical Genetics and Genomics; (-): Information not available; PM: pathogenic moderate; PP: pathogenic supporting; PS: pathogenic strong; and PVS: pathogenic very strong.

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
