# Peer review of "Comprehensive Exonic Sequencing of Known Ataxia Genes in Episodic Ataxia"

_biomedicines, 2020, doi:10.3390/biomedicines8050134_

Round 1
Reviewer 1 Report
This manuscript deals with the complex genotype-phenotype relationships in Episodic Ataxias (EAs), specifically in EA2. The authors investigate genetic association, by means of whole exome sequencing, in a modest number of undiagnosed patients from their own cohort. Their analysis focuses in genes encoding voltage-gated ion channels and transporters, and ascribes eight genes potentially linked to EA2, basically in either subgroup. Although they are mostly sodium channel subunits, there come up also potassium, chloride and calcium channels, plus a Na/K transporter and a AAA-ATPase.
The text is well written and readable. I have the following minor points:
-
Only Suppl. Table 3 and Suppl. Fig. 1 are named in the <Supplementary> File, but the main text refers to Supplementary Tables 1-4. Where are these?
-
Fig. 1 requires an explanatory legend.
-
Please verify punctuation of the sentence beginning in Line 87: “Of these 3 variants…”
-
Table 1 may need to indicate the corresponding 8 genes, according to the text (section 2.1.).
-
Sentence in Line 99 that ends with “...likely pathogenic according to the ACMG guidelines (Table 1).” may not refer to Table 1 – please, double-check. The same applies to sentence in Line 151 that reads as follows: “In total, 8 pathogenic and likely pathogenic variants were identified in the 8/16 cases (50% diagnostic yield) (Table 1).”
-
Section 2.1. lists cases but Table 1 and Supplementary Table 3 refer to ID# - please, clarify this or use a single nomenclature.
-
Re case / ID# 25, there appears to be a contradiction; text in section 2.1.2. says “migraine headache”, whereas Table 1 says “no headache” - please, clarify this.
-
In the Discussion, when commenting on ATP1A3, please change “mutations at the 756 residue” for “mutations at Arg-756”.
-
Regarding the known SCN1B mutation, C121W, which impairs disulfide bond formation between beta subunits, it is known that Scn1b null mice are ataxic and have spontaneous seizures – it could be suitable to cite also this earlier work:
Author Response
Response to Reviewer 1 Comments
We thank the Reviewers for their positive comments and the constructive suggestions to further improve our paper. We have read the reviewers’ comments and made appropriate changes. In this document, we will respond to the reviewers’ comments and detail changes to the paper in response to them.
Comments and Suggestions for Authors
This manuscript deals with the complex genotype-phenotype relationships in Episodic Ataxias (EAs), specifically in EA2. The authors investigate genetic association, by means of whole exome sequencing, in a modest number of undiagnosed patients from their own cohort. Their analysis focuses in genes encoding voltage-gated ion channels and transporters, and ascribes eight genes potentially linked to EA2, basically in either subgroup. Although they are mostly sodium channel subunits, there come up also potassium, chloride and calcium channels, plus a Na/K transporter and a AAA-ATPase.
The text is well written and readable. I have the following minor points:
We thank the reviewer for the positive appraisal of the paper.
Point 1: Only Suppl. Table 3 and Suppl. Fig. 1 are named in the <Supplementary> File, but the main text refers to Supplementary Tables 1-4. Where are these?
Response 1: We apologise for this formatting errors, we have now updated the supplementary files to include the
Supplementary Table S1
Supplementary Table S2
Supplementary Table S3
Supplementary Table S4
Supplementary Figure S1
Point 2: Fig. 1 requires an explanatory legend.
Response 2: Following the reviewer’s instructions, we have updated Figure 1 legend as follows:
Figure 1: Variant filtering workflow using Ion Reporter v5.10.
Schematic representation of the variant filtering. Tier-1 analysis (left): total variant counts at each stage of filtering using Ion Reporter software resulted in four non-synonymous pathogenic and likely pathogenic mutations in 170 ataxia-gene panel. Tier-2 analysis (right): total variant counts at each stage of filtering resulted in 7 pathogenic and likely pathogenic variants in 353 ion channel genes. All were confirmed by Sanger sequencing method.
Point 3: Please verify punctuation of the sentence beginning in Line 87: “Of these 3 variants…”
Response 3: We thank the reviewers; we have now updated this sentence to the following:
Of these, 3 variants (p.Val1325Phe in SCN2A, p.Arg756His in ATP1A3 and p.Arg167Met in KCNA1) overlap with those identified from analysis of ataxia related genes identified in Tier-1.
Point 4: Table 1 may need to indicate the corresponding 8 genes, according to the text (section 2.1.).
Response 4: We apologise for this error, Table 2 was mistakenly omitted during the upload process. We have now updated the manuscript to include Table 2 (below). Table 2 includes detailed information about the 8 genes identified and discussed in this study.
Table 2. Heterozygous mutations identified in episodic ataxia cases identified by whole exome sequencing.
ID# |
Locus (hg19) |
Ref |
Genes |
Transcript |
Amino Acid Change |
Coding |
gnomAD frequency |
ACMG rules |
verdict |
2 |
chr2:166231195 |
G |
SCN2A |
NM_001040143.1 |
p.Val1325Phe |
c.3973G>T |
- |
PM1,PM2,PP2,PP3 |
Likely Pathogenic |
6 |
chr19:42474691 |
C |
ATP1A3 |
NM_152296.4 |
p.Arg756His |
c.2267G>A |
- |
PM1,PM2,PM5,PP2,PP3,PP5 |
Pathogenic |
14 |
chr7:143048918 |
G |
CLCN1 |
NM_000083.2 |
p.Gly945fs |
c.2831dup |
0.000016 |
PVS1,PM2,PP3 |
Pathogenic |
19 |
chr6:137143923 |
C |
PEX7 |
NM_000288.3 |
p.Tyr40Ter |
c.120C>G |
0.0000907 |
PVS1, PS3, PM2, PP3. |
Pathogenic |
21 |
chr1:181689430 |
A |
CACNA1E |
NM_001205293.1 |
p.Ile614Val |
c.1840A>G |
0.00000804 |
PM1,PM2,PP2,PP3 |
Likely Pathogenic |
23 |
chr12:5021044 |
G |
KCNA1 |
NM_000217.2 |
p.Arg167Met |
c.500G>T |
- |
PM1,PM2,PP2,PP3 |
Likely Pathogenic |
25 |
chr19:35524558 |
C |
SCN1B |
NM_199037.4 |
p.Cys121Trp |
c.363C>G |
0.0000141 |
PM1,PM2,PP3,PP5 |
Likely Pathogenic |
31 |
chr2:167094721 |
A |
SCN9A |
NM_002977.3 |
p.Tyr1217Ter |
c.3651T>G |
- |
PVS1,PM2,PP3 |
Pathogenic |
Hg19: human genome 19; Ref: reference allele; ACMG: American College of Medical Genetics and Genomics; PM: pathogenic moderate; PP: pathogenic supporting; PS: pathogenic strong; PVS: pathogenic very strong.
Point 5: Sentence in Line 99 that ends with “...likely pathogenic according to the ACMG guidelines (Table 1).” may not refer to Table 1 – please, double-check. The same applies to sentence in Line 151 that reads as follows: “In total, 8 pathogenic and likely pathogenic variants were identified in the 8/16 cases (50% diagnostic yield) (Table 1).”
Response 5: We apologise for these formatting errors, and we have now corrected the errors in both sections as follows:
“Using the ataxia gene-panel list comprised of 170 genes; we identified a total of 38 novel or very rare variants. Herein, we only report the four heterozygous variants classified to be pathogenic and likely pathogenic according to the ACMG guidelines (Table 2).”
“ Using the ataxia gene-panel (Tier-1) we identified 4 causal variants in 4 cases (25%), while targeted analysis using the ion channel gene-panel list (Tier-2) identified a total of 7 variants (44%) of which 3 variants had also been identified in Tier-1. In total, 8 pathogenic and likely pathogenic variants were identified in the 8/16 cases (50% diagnostic yield) (Table 2).”
Point 6: Section 2.1. lists cases but Table 1 and Supplementary Table 3 refer to ID# - please, clarify this or use a single nomenclature.
Response 6: We have now included Table 2 which includes Case-ID number and all the identified pathogenic and Likely pathogenic mutations.
Point 7: Re case / ID# 25, there appears to be a contradiction; text in section 2.1.2. says “migraine headache”, whereas Table 1 says “no headache” - please, clarify this.
Response 7: We apologise for this accidental error, we have now updated the manuscript in section 2.1.2 as follows
“Case-25 - presented at the age of 59 with episodes of ataxia and no migraine headache. Family histories of similar symptoms were reported in his father, paternal aunt, grandmother and sisters.”
Point 8: In the Discussion, when commenting on ATP1A3, please change “mutations at the 756 residue” for “mutations at Arg-756”.
Response 8: Thank you, we have now updated the sentence as follows
“To date, including Case-6, 13 patients with mutations at the 756 residue have been reported, with symptoms including fever-induced paroxysmal weakness and encephalopathy (FIPWE).”
Point 9: Regarding the known SCN1B mutation, C121W, which impairs disulfide bond formation between beta subunits, it is known that Scn1b null mice are ataxic and have spontaneous seizures – it could be suitable to cite also this earlier work:
Response 9: We thank the reviewer for their suggestion; the reference has now been added and cited within the manuscript in the Discussion Page 7 Line 300.
“Chen and colleague reported that SCN1B null mice resulted in a dramatic phenotype that includes ataxic gait and spontaneous seizures [28].”
Reviewer 2 Report
The authors describes an extensive study started since 2016, searching for episodic ataxia mutations in known and unknown genes.
This paper report a WES analysis on 16 EA index cases resulting from previously negative screening for CACNA1A in a cohort of 31 patients with episodic ataxia. In this new screening 8 out 16 variations were identified, using two sequential steps. The results were interesting, although appear as preliminary and need to be confirmed.
Points for discussion
In Table 1 partial clinical detail and family history were reported: age of onset is often lacking (essential to define EA2 patient); ictal and non-ictal symptoms are not distinguished.
In 5 out 8 patients with identified variations, no family history were reported and one (case-31) appear sporadic case.
As Episodic Ataxia may occur sporadically or in a series of hereditary disorders, it would be interesting to know if external or acquired causes are excluded in sporadic cases, according to
Spacey S (Gene Reviews cit: Sporadic causes of episodic ataxia include multiple sclerosis, Arnold Chiari malformation, vertebral basilar insufficiency, basilar migraine, and labyrinthine abnormalities). Are these causes excluded for the six non familial cases? If not, it is possible that we search for a genetic cause where it isn’t.
Three cases appear with differential diagnosis
Case- 23 : It is not clear why this patient was included in this study. In table 1 age of onset was laking but in this patient the symptoms are typical for Episodic ataxia type 1, as discussed by the authors in line 193-194.
Case-25: appear as a non-episodic ataxia.
Case -31: appear as a progressive ataxia without episodes
Segregation study :
Case-6 and Case-25: these index cases are reported with family history. Are other family members tested for segregation? Especially for the mutation SCN1B that results as Likely Pathogenic, it was encouraged. Several authors advise segregation studies to improve mutation assessment (Amendola et al 2016 https://doi.org/10.1016/j.ajhg.2016.03.024)
Case-14 : Although there was no family history, if family members were present it would be interesting to study segregation of the CLCN1/KCNS2 variations since the same CLCN1 mutation was previously reported in a recessive disease. It would be that this index case represents a recessive form of (episodic) ataxia.
Although the results appear to be preliminary and clinical information seems rather insufficient, we appreciate this work that is, more or less, in line with other paper that try to find new genes responsible for Episodic Ataxia, by WES analysis in a selected cohort,
The authors themselves cited these variant as “suspected” and indicate that an “extensive further work” with biochemical assays, segregation and functional studies of variants …..”are needed to give a more detailed understanding and accurate classification for these variants in disease pathophysiology”
We are waiting for a more in-depth study in the future.
MINOR REVISION
Add p. before many aminoacids substitution like in
- LINE 22 and 23: add p. to Val1325Phe; p.Arg756...; p.Tyr.... p.Arg....
- LINE 93 and 94: as line 22-23
- LINE 155 and 156: as line 22-23
- LINE 160-170-173-193-268-269
LINE 29-30 This preliminary description of Episodic ataxias seems reductive for a general diagnostic classification. It is to note that myokymia is a differential characteristic of EA1. So it would be better something like:
“episodes of ataxia (discoordination) variably associated with progressive ataxia and ictal/interictal features.
LINE 38 the voltage gated calcium channel alpha1a subunit (Cav2.1) instead Voltage-Gated Calcium Channel Subunit Alpha Cav2.1,
LINE 71: the authors reports UBR4 as a “possible candidate for EA” but this gene is not present both in Tier-1 and Tier-2. Why?
LINE 87: “supplementary table S4” instead “Table 1” ?
LINE 95: in table 1 the 16 EA cases are reported but not genes and mutations. I suggest to report the mutations in table 1
otherwise cite “ table 1” near “16 cases” (line 92) and supplementary table 3 and 4 instead (Table 1) in line 95
LINE 101: (supplementary table S3 and S4) instead (Table 1)?
LINE 108: I don’t understand “post pointing”. Do you mean “past pointing” ?
LINE 153: In Table 1 mutations are not reported.
LINE 219: the VOUS reported in Supplementary Table S4 is on KCNS2 gene.
MATHERIALS AND METHODS
LINE 305: I suppose “8 primer pairs”
LINE 304/314 : I suggest to describe “variant filtering workflow” before “sanger sequencing method” , because as SS method represent a step that is performed after variant filtering.
SUPPLEMENTARY MATERIAL
Add title to the table:
- supplementary table S1 (and if you want ) tier-1 genes
- supplementary table S2
- add “S” in table 3
- supplementary table S4 variants identified in ….
Author Response
Response to Reviewer 2 Comments
We thank the Reviewers for their positive comments and the constructive suggestions to further improve our paper. We have read the reviewers’ comments and made appropriate changes. In this document, we will respond to the reviewers’ comments and detail changes to the paper in response to them.
Comments and Suggestions for Authors
The authors describes an extensive study started since 2016, searching for episodic ataxia mutations in known and unknown genes.
This paper report a WES analysis on 16 EA index cases resulting from previously negative screening for CACNA1A in a cohort of 31 patients with episodic ataxia. In this new screening 8 out 16 variations were identified, using two sequential steps. The results were interesting, although appear as preliminary and need to be confirmed.
We thank the reviewer for their evaluation of the paper. Additional experimental confirmation of variant effects may provide useful information on pathogenicity for many of the variants identified in the supplementary material, however, in the main body of the text, we have limited reporting to those variants which would be classified as pathogenic or likely pathogenic by ACMG guidelines. We believe that reporting of these variants and their clinical context is of value to the scientific and medical communities to allow comparison of symptoms and identification of these variants if they appear in other affected individuals.
Points for discussion
Point 1: In Table 1 partial clinical detail and family history were reported: age of onset is often lacking (essential to define EA2 patient); ictal and non-ictal symptoms are not distinguished.
Response 1: We understand the reviewer’s concern. Detailed clinical information including age of onset and ictal and non-ictal symptoms were not available for every patient. The 31 episodic ataxia cases were clinically evaluated and diagnosed by expert neurologists and were referred to our laboratory for EA2 molecular genetic testing before 2007. Unfortunately, for many cases, we were unable to re-contact for more extensive clinical information.
Point 2: In 5 out 8 patients with identified variations, no family history were reported and one (case-31) appear sporadic case.
As Episodic Ataxia may occur sporadically or in a series of hereditary disorders, it would be interesting to know if external or acquired causes are excluded in sporadic cases, according to Spacey S (Gene Reviews cit: Sporadic causes of episodic ataxia include multiple sclerosis, Arnold Chiari malformation, vertebral basilar insufficiency, basilar migraine, and labyrinthine abnormalities). Are these causes excluded for the six non familial cases? If not, it is possible that we search for a genetic cause where it isn’t.
Response 2: It is indeed possible that there are alternative, non-genetic, causes for the symptoms identified in our patients. Even in those with a family history, such causes might be the true culprit, and be overlooked for the more immediately obvious familial connection. Unfortunately, we do not have enough information to determine which causes have been systematically eliminated by referring clinicians, but as said above, all were diagnosed as having EA2 and referred for CACNA1A testing. Given the potential impact and expense of testing for the patient, clinicians would eliminate more likely non-genetic causes first.
This is an important point and limitation of the study, however, and as such, we have now included mention of the issue in the discussion, as follows:
“Additionally, it is possible that non-genetic or genetic and environment interacting causes may be at work in some of our patients. All patients were identified as fitting EA2 criteria well enough to warrant CACNA1A testing by their referring neurologist. Despite this, injury, infection, auto-immune diseases and similar conditions causing EA-like symptoms may have escaped clinical detection, especially if they are relatively mild and are acting in a genetic context more sensitive to ataxia related phenotypes.”
Point 3: Three cases appear with differential diagnosis
Case- 23 : It is not clear why this patient was included in this study. In table 1 age of onset was laking but in this patient the symptoms are typical for Episodic ataxia type 1, as discussed by the authors in line 193-194.
Case-25: appear as a non-episodic ataxia.
Case -31: appear as a progressive ataxia without episodes
Response 3: While the entry criteria for the study included patients being referred for EA2 testing, it also included a failure to detect a pathogenic CACNA1A variant. This immediately indicates that, whatever their phenotype, these patients do not have EA2, and the population likely encompassed several different ataxia forms. As a result, and because there is variation in phenotype even for ataxia patients with the same causative gene, we did not limit the phenotypes selected for sequencing.
Point 4: Segregation study :
Case-6 and Case-25: these index cases are reported with family history. Are other family members tested for segregation? Especially for the mutation SCN1B that results as Likely Pathogenic, it was encouraged. Several authors advise segregation studies to improve mutation assessment (Amendola et al 2016 https://doi.org/10.1016/j.ajhg.2016.03.024)
Case-14 : Although there was no family history, if family members were present it would be interesting to study segregation of the CLCN1/KCNS2 variations since the same CLCN1 mutation was previously reported in a recessive disease. It would be that this index case represents a recessive form of (episodic) ataxia.
Although the results appear to be preliminary and clinical information seems rather insufficient, we appreciate this work that is, more or less, in line with other paper that try to find new genes responsible for Episodic Ataxia, by WES analysis in a selected cohort,
The authors themselves cited these variant as “suspected” and indicate that an “extensive further work” with biochemical assays, segregation and functional studies of variants …..”are needed to give a more detailed understanding and accurate classification for these variants in disease pathophysiology”
We are waiting for a more in-depth study in the future.
Response 4: We thank the reviewer for the constructive comments. We also agree that segregation studies on these variants would add to assessment of pathogenicity. Unfortunately, as indicated in an above response, we were not able to recontact all participants for more thorough assessment, and in the absence of such additional data, we believe that the available evidence will be useful for clinicians and research teams identifying variants in their own patients and allow for additional confirmation in future
Point 5: MINOR REVISION
Add p. before many aminoacids substitution like in
LINE 22 and 23: add p. to Val1325Phe; p.Arg756...; p.Tyr.... p.Arg....
LINE 93 and 94: as line 22-23
LINE 155 and 156: as line 22-23
LINE 160-170-173-193-268-269
Response 5: We thank the reviewer; the “p.“ has now been added before every amino-acid substitution.
Point 6: LINE 29-30 This preliminary description of Episodic ataxias seems reductive for a general diagnostic classification. It is to note that myokymia is a differential characteristic of EA1. So it would be better something like:
“episodes of ataxia (discoordination) variably associated with progressive ataxia and ictal/interictal features.
Response 6: Following the reviewer’s instruction, we have now updated the sentence Line 29-30 as follows:
“Episodic ataxias are autosomal dominant neurological disorders characterised by severe episodes of ataxia (discoordination) variably associated with progressive ataxia and ictal/ and ictal/interictal features.”
Point 7: LINE 38 the voltage gated calcium channel alpha1a subunit (Cav2.1) instead Voltage-Gated Calcium Channel Subunit Alpha Cav2.1,
Response 7: Thank you, the sentence has now been updated to “the voltage gated calcium channel alpha1a subunit (Cav2.1)”
Point 8: LINE 71: the authors reports UBR4 as a “possible candidate for EA” but this gene is not present both in Tier-1 and Tier-2. Why?
Response 8: We apologise for the unintended error, we have now included the UBR4 gene variants. A total of 38 variants were identified in the UBR4 gene. Of the 38 variant, 3 non-synonymous variants were identified in the UBR4 gene in our EA2 cohort with total variants of 1457 in 170 Ataxia genes. Of the three non-synonymous variants only one classified as a variant of unknown significance (MAF<0.0001) was identified in Case-1 in the UBR4 gene (NM_020765.2 c.3077T>A p.Asn1026Ile). We have now updated the manuscript, Figure 1, supplementary Table 1 and supplementary Table 3 to include this identified gene variant.
Point 9: LINE 87: “supplementary table S4” instead “Table 1” ?
Response 9: We have now included Table 2, which includes the 8 pathogenic and likely pathogenic variants discussed in this paper.
Point 10: LINE 95: in table 1 the 16 EA cases are reported but not genes and mutations. I suggest to report the mutations in table 1
Response 10: Thank you for the suggestion; we have now included Table 2, which details the pathogenic and likely pathogenic variants. Table 2 was mistakenly omitted during the upload process.
Point 11: otherwise cite “ table 1” near “16 cases” (line 92) and supplementary table 3 and 4 instead (Table 1) in line 95
Response 11: Thank you, Table 2 has now been included to eliminate this confusion.
Point 12: LINE 101: (supplementary table S3 and S4) instead (Table 1)?
Response 12: Thank you, as indicated in an above response, Table 2 has now been included.
Point 13: LINE 108: I don’t understand “post pointing”. Do you mean “past Pointing” ?
Response 13: We have re-proofread the paper and fixed some minor typographical errors that had not been caught previously, including “post pointing” which was replaced by “past pointing”.
Point 14: LINE 153: In Table 1 mutations are not reported.
Response 14: This error has now been fixed and Table 2 now included with this information.
Point 15: LINE 219: the VOUS reported in Supplementary Table S4 is on KCNS2 gene.
Response 15: We have re-proofread the paper and fixed some minor typographical errors that had not been caught previously, including the KCNS2 gene.
MATHERIALS AND METHODS
Point 16: LINE 305: I suppose “8 primer pairs”
Response 16: Thank you, we have now replaced “16 primer pairs” with “8 primer pairs”
Point 17: LINE 304/314 : I suggest to describe “variant filtering workflow” before “sanger sequencing method” , because as SS method represent a step that is performed after variant filtering.
Response 17: Following the reviewer instruction, we have now included the variant filtering workflow section before Sanger sequencing method section on Page 9
Point 18: SUPPLEMENTARY MATERIAL
Add title to the table:
supplementary table S1 (and if you want ) tier-1 genes
supplementary table S2
add “S” in table 3
supplementary table S4 variants identified in ….
Response 18: We thank the reviewer for their suggestions; we have now updated the Supplementary Table S1, Supplementary Table S2, Supplementary Table S3 and Supplementary Table S4.